# Role of yUbp8 in Mitochondria and Hypoxia Entangles the Finding of Human Ortholog Usp22 in the Glioblastoma Pseudo-Palisade Microlayer

**DOI:** 10.3390/cells11101682

**Published:** 2022-05-19

**Authors:** Veronica De Luca, Manuela Leo, Elisabetta Cretella, Arianna Montanari, Michele Saliola, Gabriele Ciaffi, Andrea Vecchione, Antonella Stoppacciaro, Patrizia Filetici

**Affiliations:** 1Department of Biology and Biotechnologies “Charles Darwin”, Sapienza University of Rome, P.le A. Moro 5, 00185 Rome, Italy; veronica89deluca@gmail.com (V.D.L.); manuelaleo10@gmail.com (M.L.); elycretella@gmail.com (E.C.); ari.montanari@uniroma1.it (A.M.); michele.saliola@uniroma1.it (M.S.); 2Department of Clinical and Molecular Medicine, Sant’ Andrea Hospital, Sapienza University of Rome, P.le A. Moro 5, 00185 Rome, Italy; ciaffi.1678724@studenti.uniroma1.it (G.C.); andrea.vecchione@uniroma1.it (A.V.); 3Institute of Molecular Biology and Pathology—CNR, Sapienza University of Rome, P.le A. Moro 5, 00185 Rome, Italy

**Keywords:** Gcn5, Ubp8, Usp22, mitochondria, hypoxia, glioblastoma, pseudo-palisade

## Abstract

KAT Gcn5 and DUB Ubp8 are required for respiration and mitochondria functions in budding yeast, and in this study we show that loss of respiratory activity is acquired over time. Interestingly, we show that absence of Ubp8 allows cells to grow in hypoxic conditions with altered mitophagy. Comparatively, the aggressive glioblastoma (GBM) multiforme tumor shows survival mechanisms able to overcome hypoxia in the brain. Starting from yeast and our findings on the role of Ubp8 in hypoxia, we extended our analysis to the human ortholog and signature cancer gene Usp22 in glioblastoma tumor specimens. Here we demonstrate that Usp22 is localized and overexpressed in the pseudo-palisade tissue around the necrotic area of the tumor. In addition, Usp22 colocalizes with the mitophagy marker Parkin, indicating a link with mitochondria function in GBM. Collectively, this evidence suggests that altered expression of Usp22 might provide a way for tumor cells to survive in hypoxic conditions, allowing the escape of cells from the necrotic area toward vascularized tissues. Collectively, our experimental data suggest a model for a possible mechanism of uncontrolled proliferation and invasion in glioblastoma.

## 1. Introduction

Mitochondrial disorders affect at least 1/5000 newborns and are caused by mutations in nuclear or mitochondrial DNA (mtDNA) [1]. Defects in mtDNA may result in pathological alteration of respiration and energy metabolism and often correlate with cancer, neurodegenerative and cardiovascular diseases [2]. Aging has been associated with decline in mitochondrial quality and activity [3]. The study of molecular mechanisms and factors involved in mitochondrial regulation are considerably facilitated in model systems like yeast due to its fermentative metabolism and tolerance to mutations altering oxidative phosphorylation, or loss of mitochondrial functions [4]. Mitochondria are autonomous organelles devoted to metabolic processes expressing their own mitochondrial products including factors involved in oxidative phosphorylation [5] for energy and ROS production [6]. In the case of mitochondrial dysfunction the yeast Retrograde Response (RR) is activated for adaptation to defective respiration and the expression of RR genes is upregulated before induction of mitophagy [7,8]. The multiprotein complex SAGA/SLIK, composed of Gcn5, Rtg2 and Ubp8 is directly involved in RR and in life span control [9,10]. In eukaryotic cells, major cellular functions are preserved by mitochondrial homeostasis with mechanisms such as mitophagy [11,12,13], a selective form of autophagy leading to the degradation of defective mitochondria during adaptation of cells to nutrients or removal of toxins and damaged macromolecules [14,15,16,17]. The induction of autophagy in yeast is altered if mitochondrial respiration is compromised [18,19]. The key mitophagy machinery is also regulated by extensive post-translational modifications (PTMs) [20] like ubiquitin dependent pathways involving Ubp3 and Ubp16 [21,22]. Acetylation can also modulate mitophagy through GCN5L1, human ortholog of Gcn5, involved in the acetylation of mt-proteins [23,24,25,26,27]. Consistently, we reported the role of KAT-Gcn5 and ubiquitin protease Ubp8 in yeast respiration and their localization into mitochondria as single proteins [28,29,30]. Glioblastoma multiforme (GBM) is one of the most aggressive brain tumors, resistant to therapy and with a high grade of invasiveness. It is distinguished from lower-grade gliomas by specific histological hallmarks like pseudo-palisade tissue surrounding a more or less extended necrosis area, its microvascular proliferation makes GBM one of the most hypoxic and angiogenic tumors [31]. The organization of GBM in layers with distinct microenvironmental niches supports the tumor’s metabolic needs, survival, proliferation and stem cell maintenance. The hypoxic GBM niche is characterized by the pseudo-palisade necrosis area that recruits innate immune cells including macrophages that promote cancer stem cells’ (CSC) proliferation and expression of hypoxia inducible factor alpha1 and 2 (HIF-1α and HIF-2α) [32,33]. Hypoxia generally occurs when tumor growth exceeds neovascularization. The perivascular GBM niche (PVN) is composed of non-neoplastic cells and glioma interacting cells providing a supportive micro-environment for cancer stem cell (CSC) growth, maintenance, and survival [34]. Finally, the invasive GBM niche is localized at the outer edge where glioma cells invade and migrate along blood vessels. Pseudo-palisading necrosis and microvascular hyperplasia are predictors of poor prognosis among diffuse gliomas and distinguish the transition from a high-grade astrocytoma to glioblastoma, supporting the functional involvement of hypoxic and necrotic regions in tumor progression and aggressiveness [35]. In this study we have analyzed the importance of the ubiquitin protease Ubp8 and acetyltransferase Gcn5 in a yeast model, showing that mitochondrial defects are acquired over time, and altered expression of Ubp8 induces defects in mitophagy thus allowing an unexpected capacity for growth during hypoxia. Starting from these findings and previously published data, we extended our analysis to the human ortholog Usp22, one of the signature molecules of aggressive glioblastoma [36,37]. We found that Usp22 is overexpressed in tumor tissues and demonstrate that it is selectively overexpressed in the GBM pseudo-palisade micro layer around the central necrotic area. This result along with the data obtained in yeast, suggest that altered expression of Usp22 might be involved in mitochondrial functions allowing the survival of tumor cells in low oxygen. Our collected results may therefore provide a model where Usp22 localized at the edge of the tumor necrotic area may regulate uncontrolled proliferation of tumor cells at the edge of the pseudo-palisade, boosting the survival of cells able to proliferate and migrate toward the outer oxygenized tissues and blood vessels.

## 2. Materials and Methods

### 2.1. Yeast Strains, Growth

Yeast strains used are listed in Table 1. Gene disruption of GCN5 and UBP8 was carried out with an integrative marker cassette and controlled by colony PCR. Strains named “New” were disrupted de novo, “Old” YPO4 (*gcn5*Δ-O) and YFT21 (*ubp8*Δ-O) strains were routinely used in the laboratory for years. The W303-rho^0^ strain was obtained with ethidium bromide treatment. Cells were grown at 28 °C, in YP medium (1% yeast extract, 2% bactopeptone) containing 2% glucose. Growth spot assay was performed in exponentially growing cells (1 OD600/mL), serially diluted (1:5) and spotted on agar-YP medium containing 2%, 0.2% glucose, 3% glycerol. Antimycin-A (2 μM) was added for blocking the respiratory chain. Hypoxia was generated in tightly capped anaerobic jars with the Anaero-Gen gas pack (OXOID Anaero-Gen).

### 2.2. O_2_ Consumption Measurement

Respiration studies were performed using a Clark oxygen electrode Hansatech Instruments as previously described [38]. Yeast cells (0.03 g/wet weight) grown in 0.25% glucose YP medium were collected, washed with sterile water and resuspended in 1 mL of sodium-phosphate buffer (10 mM, pH 7.4), plus glucose (20 mM). Samples were loaded into the reaction vessel of an oxygen electrode chamber.

### 2.3. Total Protein Extraction, SDS-PAGE, Western Blot and Immunohistochemistry

Cells grown at 28 °C in 2% glucose medium were collected at exponential phase (0.8 OD600/mL). Cell pellets were treated with 0.3 M NaOH, 140 mM βME, incubated on ice, TCA (55%) was added with a further incubation on ice. Cell pellets were lysed in HU-buffer (8 M urea, 5% SDS, 200 mM Tris-HCl: pH 6.8, 0.1 mM EDTA, 100 mM DTT, bromophenol-blue) at 65 °C for 15 min. Protein extracts were loaded on 10% SDS–PAGE, electrophoresed, blotted on nitrocellulose membranes (Amersham Biosciences, Amersham, UK) and hybridized with anti-Porin1 (Invitrogen, Waltham, MA, USA) and anti-Ada2 (Santa-Cruz, Dallas, TX, USA). Proteins were detected by SuperSignal West Dura Extended Duration Substrate (Thermo Fisher Scientific, Waltham, MA, USA) and visualized by a ChemiDoc™ MP Imaging System (Biorad, Hercules, CA, USA), signals were quantified by Image-Lab analysis software (Version 5.2.1, Bio-Rad Laboratories, Inc., Hercules, CA, USA).

### 2.4. Rosella Experiment

Strains were transformed with Rosella plasmid pAS1NBm-RosellaI [39], Ros01, Ros02, and Ros03 strains are listed in Table 1. Newly disrupted strains containing the Rosella plasmid Ros01 and Ros03 were allowed to grow in NSC medium (2% glucose, 0.67% yeast nitrogen base with ammonium sulfate plus amino acid mix) by repeated dilution of cultures at exponential log phase for several days until approximately 60 generations (60 G). Visualization of the mitochondrial network was performed with exponentially growing cells at log (OD 0.6–0.8) and stationary (2.2–4) phases. Images were acquired with a Nikon Ti2 microscope using X-light V3 spinning disc confocal (Crest-Optics, Rome, Italy) and controlled with NIS Elements software (Nikon, Tokyo, Japan).

### 2.5. Fluorescent Imaging

Strains were transformed with pVT100U-mtGFP plasmid (GFP-5′ATPase-9, Westermann B., 2000), YVDL09, YVDL11, YVDL12, YVDL13, YVDL14 and YVDL15 strains are listed in Table 1. Visualization of active mitochondrial membranes was performed on exponentially growing cells supplemented with 1:1 DASPMI [2-(4-(dimethylamino steryl)-1-methylpyridinium iodide] (100 μM) on glass slides. Active mitochondria stained with DASPMI and strains transformed with mtGFP were visualized on a Zeiss Axio Imager Z1 Fluorescence Microscope, images were acquired by an Axio-Vision 4.8 Digital Image Processing System and objective lens 63× with oil.

### 2.6. Real Time qRT-PCR and End-Point PCR

Exponentially growing cells (WT, *gcn5*Δ and *ubp8*Δ strains) were collected after overnight growth in YP glu-2%. Total RNA and DNA were extracted with the phenol method; total RNA was retro-transcribed with Quanti-Tect Reverse Transcription Kit (Qiagen, Hilden, Germany). Standard curves of WT genomic DNA (20/0.05 ng) and cDNA (50 ng) were amplified for Actin, POR1 and OXI1 with primers (listed). All samples were measured in at least three independent experiments and normalized to ACT1 mRNA. Quantitative RT-PCR (qRT-PCR) experiments were carried out on a Rotor-gene Q (Qiagen, Hilden, Germany) apparatus. End-point PCR was performed using Applied Biosystem Thermocycler to amplify genomic DNA with OXI1 and ACT1 specific primers (listed). qRT-PCR and End-Point PCR products were run on a 2% agarose gel.
ACT1 Fw: 5′-GCTGAAAGAGAAATTGTCCG-3′;ACT1 Rev: 5′-ACACTTCATGATGGAGTTGTA-3′;OXI1 Fw: 5′-GTACCAACACCTTATGCAT-3′;OXI1 Rev: 5′-CATTCAAGATACTAAACCTAA-3′;POR1 Fw: 5′-CAAGGATTTCTATCATGCTACC-3′;POR1 Rev: 5′GCTTGTCATTCAACTTTGCTTC-3′.

### 2.7. Statistical Analysis

Each experiment represents the average of at least three different biological replicates unless otherwise stated in the figure legend. Standard errors are shown, and asterisks are as follows: **** *p* < 0.0001, *** *p* < 0.001, ** *p* < 0.01, * *p* < 0.05. Statistical analysis was performed with the Student’s *t*-test.

### 2.8. Glioma Samples

Sequential 4 mm sections were obtained from paraffin blocks of thirty glioblastomas chosen among the 83 glioblastomas stored in the histoteque of the Pathology Unit of Sant’Andrea University Hospital. The samples were obtained from the Department of Neurosurgery for diagnostic purposes between 2019 and 2021. All procedures related to the use of the samples were consented by the patients and the study approved by the ethics committee. The diagnosis was made following the WHO 2016 classification [40]. Stereotactic biopsy and largely necrotic samples were excluded from the study.

### 2.9. Immunohistochemistry

Sections were deparaffinized in xylene and hydrated in a series of graded ethanol. Antigen retrieval was performed using DAKO PT-link pH 6 or pH 9 for 20 min a 98 °C; endogenous peroxidase was inhibited by 0.3% hydrogen peroxide for 10 min. Sections were then incubated 1 h at RT with anti-USP22 (1:200, retrieval pH6, Abcam, Cambridge, UK), -Parkin (1:100 retrieval pH9, Cell Signaling Technology, Danvers, MA, USA) and Ki-67 (1:200 retrieval pH6, Dako, Camarillo, CA, USA), washed in 1 × PBS pH 7.4 and re-incubated for 30 min at RT with Dako EnVision^®^ (Dako, Camarillo, CA, USA) + Dual Link System-HRP (DAB+) and finally counterstained with Hematoxylin.

Percentages of positive tumor cells scored 0 (0% to 25%), 1 (26% to 50%), 2 (51% to 75%), and 3 (76% to 100%), staining intensity was graded as follows: 0 (negative), 1 (weak), 2 (moderate), or 3 (strong). The arithmetic sum of the two scores are considered for statistical purposes.

## 3. Results

### 3.1. Lack of Gcn5 and Ubp8 Induces a Progressive Loss of Respiration in Time

Our previous studies showed that Gcn5 and Ubp8 are required for respiration in *S. cerevisiae* [28,29]. In this study we wanted to understand if the loss of mitochondrial function was due to disruption of genes or depended on time. To obtain this information we performed a de novo disruption of GCN5 and UBP8 and obtained *gcn5*Δ-N (New) and *ubp8*Δ-N strains. Growth in fermentative glu-2% and respiratory gly-3% of these strains (Figure 1A) was compared with the fully respiratory WT (rho^+^) strain and the corresponding WT (rho^0^) where mtDNA was eliminated by EtBr treatment (see material and methods). In respiratory medium, gly-3%, either *gcn5*Δ-N and *ubp8*Δ-N were able to grow. Conversely and according to previous reports, the old *gcn5*Δ-O and *ubp8*Δ-O strains were defective and showed poor growth in respiratory conditions. Collectively these experimental data confirm that respiratory inability is not dependent on the gene disruption per se, but occurs over time. We then measured the O_2_ consumption rate of *ubp8*Δ and *gcn5*Δ new and old strains compared with wild type rho^+^ and rho^0^ (Figure 1B). Upper panel shows O_2_ consumption of *ubp8*Δ-N (red), and *ubp8*Δ-O (yellow), lower panel the same analysis performed on the indicated strains disrupted in GCN5. Collected results confirm the growth spot assay and indicate a very clear decrease in oxygen utilization in the old strains, suggesting that loss of respiratory capacity is acquired during time. The mitochondrial network is generated by a continuous dynamic process of fusion and fission of mitochondrial membranes in response to the metabolic necessity of the cell [41]. Moreover, its dysfunction correlates with mitochondria degradation recurrent in several human neurodegenerative disorders.

We therefore performed a microscopic analysis of the mitochondrial morphology stained with Mito-GFP [42]. Figure 1C shows that the tubular mitochondrial network is evenly distributed in WT rho^+^ cells, while it is lost in WT rho^0^ lacking mitochondria, that display a punctuated and fragmented mitochondrial network. Similarly, to WT a well-shaped mt-network was found in the newly disrupted *gcn5*Δ-N and *ubp8*Δ-N strains whereas in the older *gcn5*Δ-O and *ubp8*Δ-O it was fragmented and the widespread presence of fluorescent dots suggested an unhealthy state of mitochondria. The functionality of mitochondrial membranes was also controlled by staining with DASPMI, a specific dye that binds to mt-membranes with an active Δψ potential. Similarly, to the results obtained with mtGFP (Figure 1C) we found low staining and loss of the tubular network in *gcn5*Δ-O and *ubp8*Δ-O strains, while the N-strains showed a well-organized and brilliant tubular network. These data suggest again, that the membrane potential is altered and, in general, that the mitochondrial network is absent and heavily altered in Old strains disrupted in GCN5 or UBP8.

### 3.2. Mitochondria Components Are Lost in gcn5Δ and ubp8Δ Old Strains

We then asked whether the observed decrease in respiration in old strains were correlated to loss of mitochondria or mt-components. Since the respiratory capabilities are associated with mtDNA, we checked for its presence by amplification of OXI1, a mitochondrial gene coding for the subunit II of the cytochrome C oxidase [43] in the different strains. End point amplification of OXI1 (Figure 2A) showed an almost complete loss in *gcn5*Δ-O and *ubp8*Δ-O when compared to the actin gene (ACT1), similar to a Wt rho^0^ strain deprived of mtDNA. qRT-PCR performed on the same strains confirmed the drastic loss of mtDNA in the absence of Gcn5 and Ubp8 (Figure 2A). Porin1 (Por1) is a protein localized on the outer membrane of mitochondria that regulates permeability. We decided to analyze Por1 in order to monitor the degradation of mitochondrial proteins occurring in mitophagy, as previously described [44]. We prepared total protein extracts from WT, rho^+^ and rho^0^, *gcn5*Δ and *ubp8*Δ, respectively, New (N), and Old (O) strains that were hybridized in a western blot with anti-Por1 and anti-Ada2 (Figure 2B). Por1 abundance was normalized with Ada2 and the histogram showed conclusive results. In rho^0^ strain there is a reduced amount of Por1, suggesting that, despite the loss of mtDNA, rho^0^ cells still require maintenance of the mitochondrial structure. Notably, the extremely low levels of Por1 in the Old, *gcn5*Δ-O and *ubp8*Δ-O strains, indicated that, along with mtDNA loss, impairment of mitochondrial functions were probably associated with loss of mitochondrial mass and inefficient membrane dynamics causing the observed fragmentation of the mitochondrial tubular network. We then asked whether the decrease in Porin 1 might be also due to its lowered mRNA expression. For this, we analyzed POR1 mRNA. Figure 2C shows that transcription of POR1 doesn’t decrease in *gcn5*Δ-O and *ubp8*Δ-O strains with respect to wild type, ruling out a transcriptional effect affecting Porin 1 decrease. Collectively, these experimental data demonstrate that there is a consistent and progressive loss of mitochondrial proteins in elderly strains that fits with respiratory defects and aberrant morphology of mitochondria. Noteworthy, it was reported that Porin1 has a critical function in mitochondrial phospholipid metabolism and is required for phospholipid transport from the endoplasmic reticulum to the outer membrane of mitochondria [45]. In particular, when yeast cells undergo transition from fermentative to respiratory conditions the phospholipid transport becomes responsible for about 40–50% of the cellular phosphatidylethanolamine in mitochondria [46] which is required for respiratory activity and mitochondrial quality control. In the light of these additional features, the loss of Porin1 observed in, respectively, *gcn5*Δ-O and *ubp8*Δ-O strains is even more effective on mitochondrial activity.

### 3.3. Yeast Cells Missing Ubp8 Are Able to Grow in Hypoxic Conditions and Show Abnormal Mitophagy

Gcn5 and Ubp8 are needed in respiration [28,29] and their loss induces an altered utilization of glucose with defects in glycolysis [47]. The data showing the progressive loss of mitochondria over time occurring in *gcn5*Δ and *ubp8*Δ old strains may therefore affect not only mitochondrial functions but also the overall metabolic response of cells. These findings suggest that an altered utilization of metabolic routes branching out of the central glycolytic pathway may occur, which are able to provide the intermediates necessary for the replenishment of anaplerotic routes. To test if other stress-induced pathways might be affected by loss of respiratory function we challenged the fermentative growth by inducing hypoxia, and a block of respiratory complex III with antimycin-A. Collected results show that in the absence of Gcn5, strains were fully dependent on the mitochondrial functionality, accordingly, the *gcn5*Δ-O strain was unable to grow in hypoxia. Conversely, the growth of *ubp8*Δ-O was unaffected by an absence of O_2_ (Figure 3A) and growth in hypoxia was confirmed by efficient growth in the presence of the respiration blocking antimycin-A. These results highlight the unexpected capability to survive in conditions of low oxygen by activation of alternative metabolic routes in the absence of Ubp8. The degradation of mitochondria by mitophagy is of particular importance for preserving the active mitochondrial pool in cells and their turnover. We therefore asked whether the feature of survival in hypoxic conditions in the absence of Ubp8 might also induce defects in mitophagy. In order to investigate this point we used the fluorescent reporter Rosella, a dual-emission biosensor expressing a pH-stable red fluorescent protein fused to a pH-sensitive Green Fluorescent Protein variant [39]. The rationale of the experiment is shown in the scheme of Figure 3B, the expressed fluorescence depends on the pH of the environment in the vacuole and in other compartments. The red fluorescent protein is quite stable in the vacuole at pH 6.2 or lower, while the GFP is almost negligible, while in the cytosol or other compartments, pH 7.0, red and green proteins are evenly expressed. Therefore, in log phase growing cells the red and green fluorescent proteins are co-expressed, distributed in the cytosol but absent from the vacuole. In stationary phase, starvation autophagy is induced, and only red fluorescence is expressed and accumulated in the vacuoles. Rosella was introduced in wild type and in the newly disrupted *ubp8*Δ-N strains in order to follow the onset of mitophagy. Strains were maintained under continuous growth conditions by repeated dilution of cultures at early log phase for several days up to, approximately, 60 generations. Fluorescence of cells was followed with confocal microscopy in cultures harvested at early log and late stationary phases. Figure 3C shows the collected results. In log phase WT and *ubp8*Δ cells showed the typical distribution of yeast mitochondrial reticulum stained in red/green. In the log/stationary phase the vacuoles of WT strain appeared stained red indicating the accumulation of the Rosella red fluorescent proteins inside vacuoles. Interestingly, the vacuolar localization of Rosella was less defined in *ubp8*Δ cells meaning that mitophagy was not so efficiently activated in the cells. Overall, this experiment suggests that mitophagy is altered in absence of Ubp8 with respect to wild type.

### 3.4. The Human Ortholog of Ubp8 Usp22 Is Overexpressed in the Pseudo-Palisade Tissue of Glioblastoma Multiforme Tumor Specimen

Based on the high evolutionary conservation from yeast to humans we wanted to analyze the localization of Usp22, the ortholog of Ubp8, which is a cancer signature gene overexpressed in aggressive GBM with poor prognosis [36]. In particular, we were interested to investigate whether Usp22 expression was differently modulated between well oxygenated and hypoxic tumor areas. In fact, a prominent feature of GBM is the presence of specialized tumor niches characterized by a central hypoxic necrosis area surrounded by a row of neoplastic elongated cells disposed to form a palisade organized in subsequent layers; this peculiar morphology is representative of the actual survival and migration of the neoplastic cells toward vascularized areas of glioblastoma usually localized on the periphery of the palisades [31]. Sections from 30 different glioblastomas were immunostained with anti Usp22 and the reactivity evaluated as a sum between staining intensity and percentage of positive cells. As reported in Table 2 palisade cells were always positive. Interestingly, in 19 of 30 cases all the palisade areas present in the sections showed a significant increase in Usp22 expression when compared with the oxygenated areas of the same tumor (Figure 4A). In order to understand if Usp22 similarly to Ubp8, might have links with mitochondria, we then immunostained tumor specimens with anti-Parkin. Parkin is recruited in mitochondria as the first step of mitochondrial quality control in response to mitochondrial damage [48,49]. Our results indicate that while some GB cells show a faint diffuse cytoplasmic staining, in the cells of the pseudo-palisade the stain signal is quite strong (Figure 4B) indicating the recruitment of Parkin at mitochondria and activation of mitophagy in the pseudo-palisade. Finally, to understand whether the pseudo-palisade cells were able to proliferate sequential sections were immunostained with anti-Ki67 antibody. As shown in Figure 4C, most of the Usp22/Parkin positive palisade cells closely surrounding the necrosis were Ki67 negative and the frequency of Ki67 positive nuclei was higher in the more external layers, thus suggesting that palisade structures provide a proliferative advantage to the tumor cells possibly due to overexpression of Usp22 and induction of mitophagy. Collectively our findings indicate that Parkin and Usp22 colocalize in the anoxic pseudo-palisade tissue of GBM and reveal a functional link between Usp22 and the activation of mitochondria quality control in the pseudo-palisade hypoxic tissue. A matter to be investigated in more detail is how the overexpression of Usp22 may induce survival in the poor oxygen and high sugar environment and whether these features may recall the yeast phenotype described in absence of Ubp8.

## 4. Discussion

We reported that the subunits of the SAGA complex [50] Gcn5 and Ubp8, besides their nuclear role of chromatin remodeling, are required for yeast respiration and are localized, as single independent proteins, into mitochondria [28,29,30]. This evidence suggests an additional, important role of these functions in metabolism and mitochondrial function. Starting from these findings, here we show that loss of respiration is acquired over time and doesn’t depend on the disruption of GCN5 and UBP per se. Accordingly, we found that the ability of *gcn5*Δ and *ubp8*Δ strains to grow on respiratory carbon sources was progressively lost from new (N) to elderly (O) strains indicating a continuous and progressive loss of mitochondrial functions. Mitochondrial damage was also demonstrated by the fragmented and punctuated mitochondrial tubular network found in the Old *gcn5*Δ and *ubp8*Δ strains. Loss of mt-DNA and mt-proteins was also observed in the absence of Gcn5 and Ubp8. Interestingly, the effects of Gcn5 and Ubp8 on yeast growth were different. In fact, while loss of Ubp8 allowed the yeast strain to grow in hypoxic conditions and under blocked respiration in the presence of antimycin A, disruption of Gcn5 was ineffective and cells didn’t grow in an absence of O_2_. In short, these results suggest that lack of Ubp8, in contrast to Gcn5, enabled yeast cells to grow in strictly anaerobic conditions which lead to higher glycolytic fluxes and enhanced fermentative capabilities, as previously reported [47]. Starting from our collected results showing the important role of Ubp8 in respiratory functions and mitophagy in budding yeast we decided to analyze whether Usp22–its human ortholog, that is an important biomarker and a cancer signature gene in aggressive glioblastoma–might show any specific localization in tumor specimens of glioblastoma tissues. Histopathologic analysis of GBM tumor specimens allowed the localization and visualization of the overexpression of Usp22 in the pseudo-palisade tissue around the central necrosis area of the tumor. Glioblastoma has anoxic tissues surrounded by high concentrations of sugar; in this regard its environment is similar to the hypoxic, rich sugar conditions tested in yeast for the *ubp8*Δ strain. We suggest that the Usp22 overexpression found in GBM might be compared to Ubp8 loss in yeast since both cases underline an altered expression of Ubp8 and Usp22 that affect the ability of cells to grow in the absence of O_2_.

The localization and overexpression of Usp22 in this tissue, never reported before, underline a respiratory function of Usp22 that is specifically exerted on the basis of its localization in a hypoxic environment. In this context, in our proposed model, the peculiar localization of Usp22 in the pseudo-palisade of GBM tissue around the necrosis area indicates that tumor cells survive the anoxic conditions, proliferate and then are able to migrate outside the pseudo-palisade toward vascularized tissue to finally achieve uncontrolled proliferation into vessels.

In this study we have used the simple yeast as a model to uncover novel features in higher eukaryotes linked to loss of the epigenetic factors Gcn5 and Ubp8. Strikingly, our results prompted us to analyze glioblastoma tumor tissues where Usp22 has been reported to be a cancer signature linked to tumor prognosis. Although preliminary, this study suggests an interesting convergence of functions between Ubp8 and Usp22, both involved in the activation of pathways for the survival of cells in anoxic situations. We cannot exclude, if the role of Usp22 in the response to anoxic conditions is confirmed, that specific inhibitors for this epigenetic factor may help to control GBM.

The first report of the cellular localization of Usp22 in the pseudo-palisade tissue and its colocalization with the mitophagy marker Parkin, which is recruited to mitochondria in case of respiratory defects and mitochondrial misfunctioning, is in addition remarkable and once more indicates the relationship between Usp22 and the capability of glioblastoma cells to survive in hypoxic conditions. Further studies are required in order to test in cell lines how overexpression of Usp22 allows cells to survive hypoxia and proliferate into vessels and outer tissues. We believe that this model may represent a first attempt for future research to uncover the mechanism and metabolic processes involved in glioblastoma and mitochondria functions linked to the cancer gene Usp22.

## 5. Conclusions

The role of epigenetic factors and the cross talk between post-translational modifications in the regulation of mitochondrial functions and respiration in yeast can be paradigmatic for the study of cancer metabolism where alteration of the microenvironment and hypoxia are recurrent in solid tumors [51,52]. Noteworthy, the ortholog of the yeast Ubp8 and cancer signature gene Usp22 [53] is a hallmark of aggressive glioblastoma tumors. The findings describing the role of Ubp8 in mitochondrial respiration and the effects of its disruption in inducing defective mitophagy and capability to grow in the absence of oxygen, open future investigations focusing on convergent functions of Usp22. Especially considering our data demonstrating its selective localization in the pseudo-palisade GBM layer around the central necrosis. Starting with our data in yeast and considering the high evolutionary conservation between Ubp8 and Usp22 [53], we would like to propose a working model for future experimental approaches aimed at investigating if aberrant expression of Usp22 might affect the anaerobic growth of specific cells allowing their uncontrolled proliferation in cancer.

## Figures and Tables

**Figure 1 cells-11-01682-f001:**
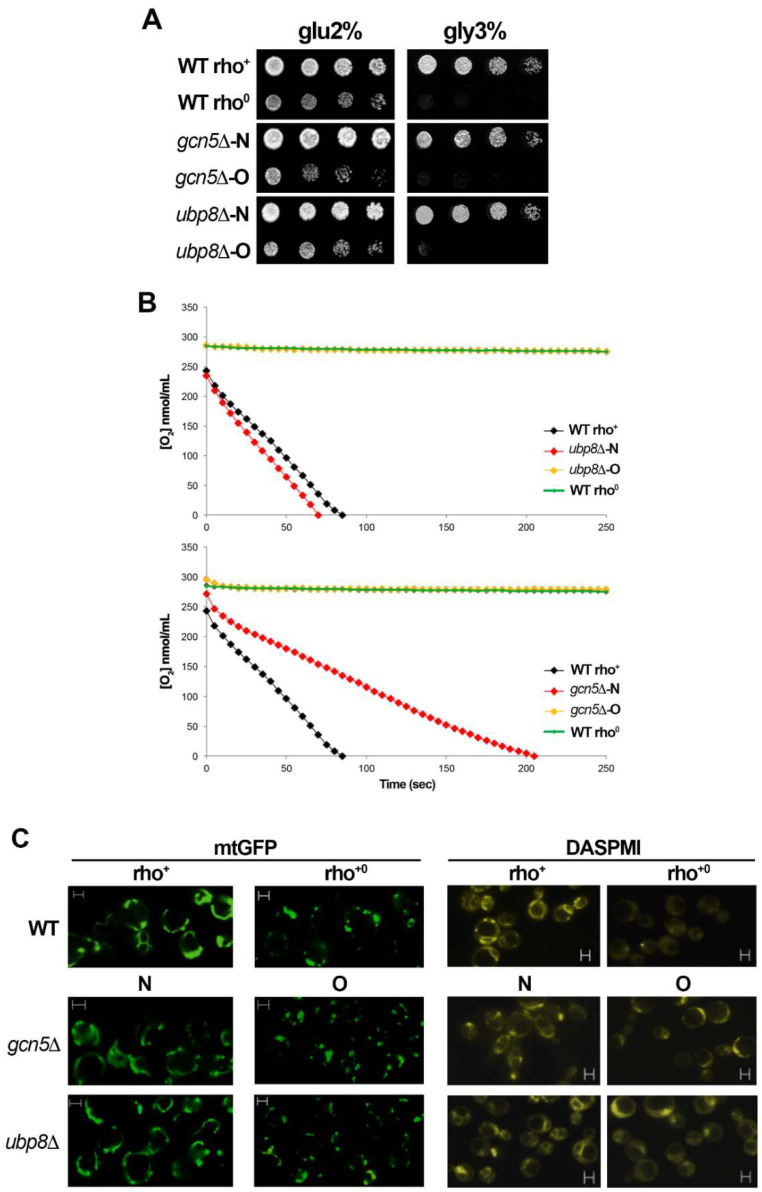
Disruption of GCN5 and UBP8 causes loss of respiratory functions in elderly strains. (**A**) Growth in YP glu-2% and gly-3% was performed on serially diluted yeast strains. WT rho^+^ and rho^0^ were compared with *gcn5*Δ and *ubp8*Δ newly disrupted (N) and old (O) strains. (**B**) Oxygen consumption rate, expressed as O2 nmol/mL of WT rho^+^ (black), WT rho^0^ (green), *ubp8*Δ-N (red) and *ubp8*Δ-O (yellow) and the same series disrupted in GCN5. Strains were grown in YP glu-0.25% medium. (**C**) Mitochondria were stained and visualized in fluorescent microscopy with mt-GFP and DASPMI, WT rho^+^ and rho^0^, gcn5Δ and ubp8Δ, respectively, New (N) and Old (O). Scale Bar (2 μm).

**Figure 2 cells-11-01682-f002:**
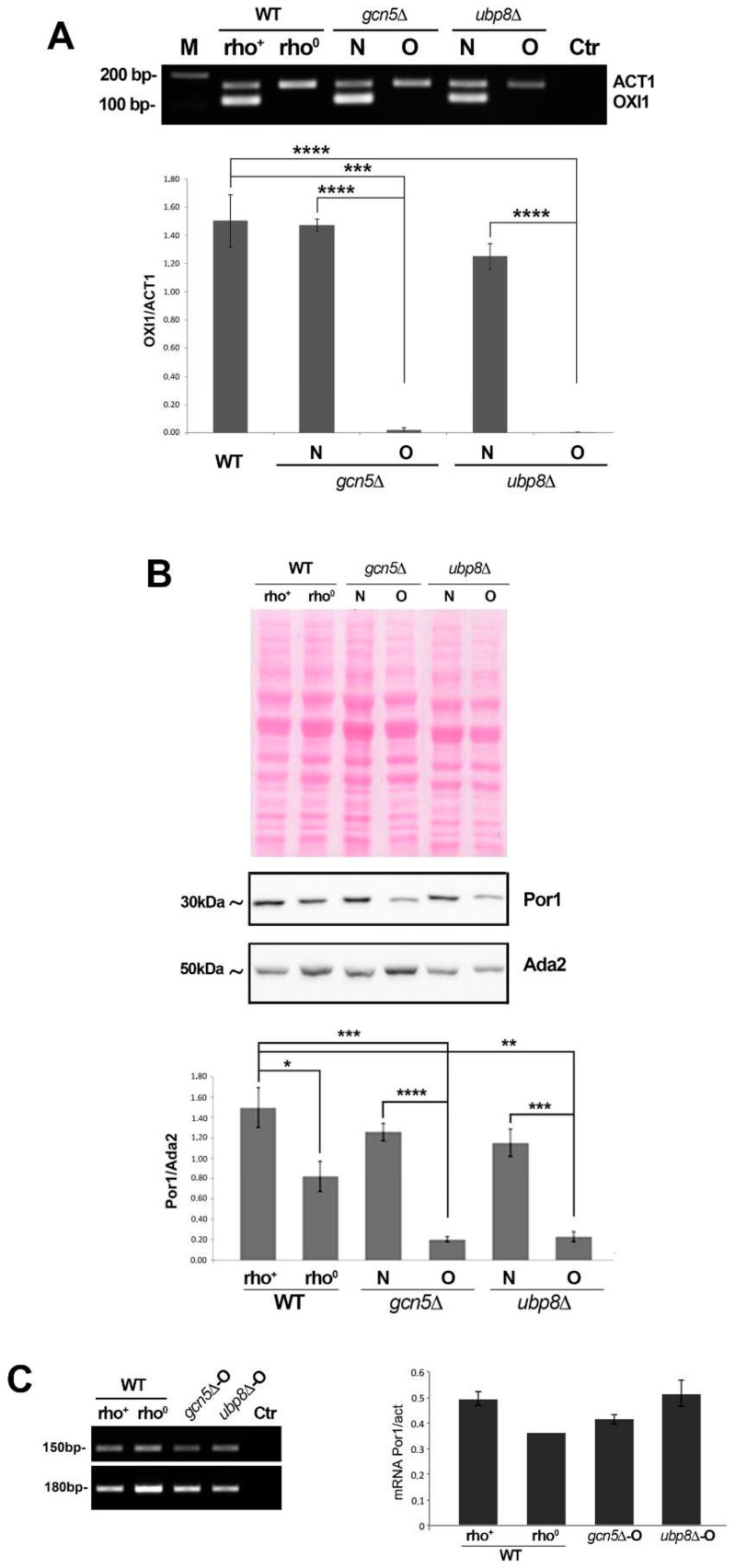
Mitochondria components are lost in gcn5Δ and ubp8Δ Old strains. (**A**) End point PCR from total DNA preparations of actin and OXI1 in the indicated strains and qRT-PCR results of OXI1-DNA normalized to actin in the indicated strains. (**B**) Western blot analysis of total protein extracts of indicated strains grown in YP glu-2% sequentially hybridized with anti-Por1 and anti-Ada2 antibodies. Histogram shows the relative abundance of Por1p/Ada2p in the indicated strains. Standard errors are shown, and asterisks are as follows: **** *p* < 0.0001, *** *p* < 0.001, ** *p* < 0.01, * *p* < 0.05. Statistical analysis was performed with the Student’s *t*-test. (**C**) End point amplification of POR1 and actin mRNA and qRT-PCR analysis of mRNA expression of POR1/actin in the indicated strains.

**Figure 3 cells-11-01682-f003:**
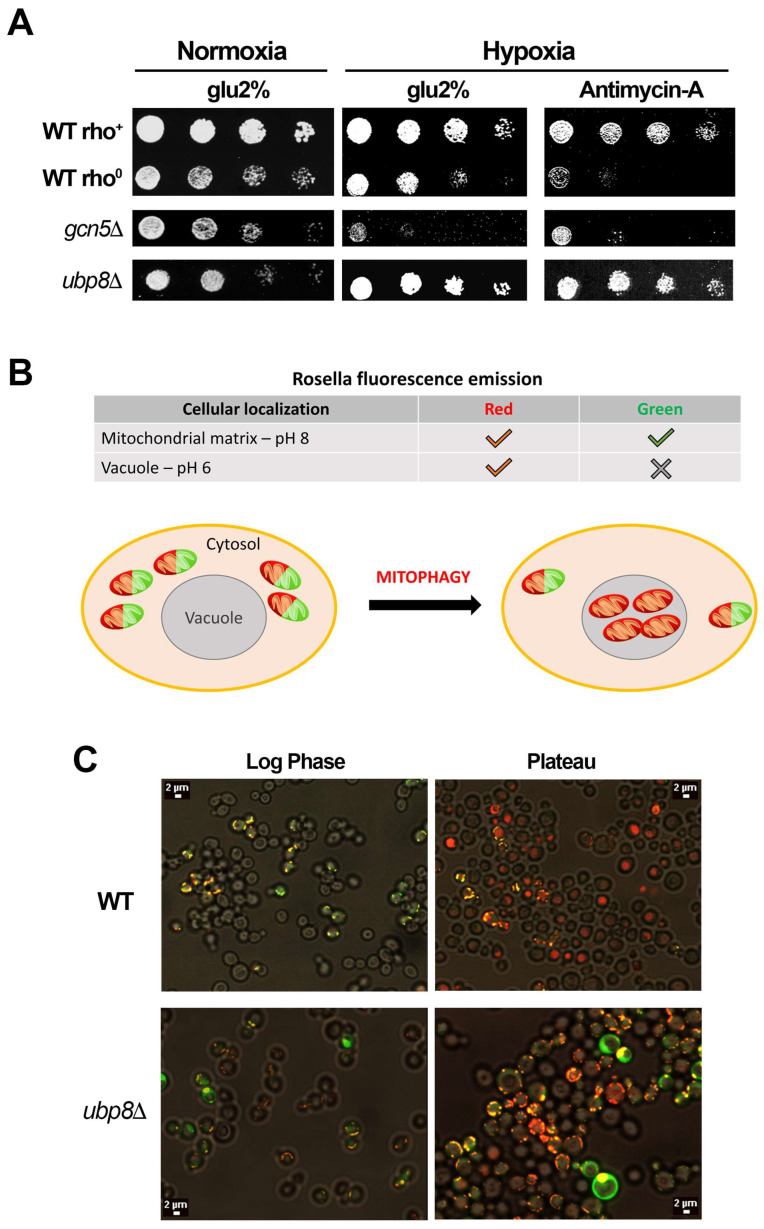
Hypoxia and mitophagy in the absence of Gcn5 or Ubp8. (**A**) Serial dilutions of indicated strains grown in YP glu-2% normoxia (+O_2_), hypoxia (−O_2_) and plus antimycin-A (2 mM). (**B**) Schematic representation of mitochondrial Rosella reporter expressing GFP (green) exclusively at pH ≥ 7.0, red fluorescence at pH < 7.0. √ = Found; × = Not Found. (**C**) Fluorescent microscopy of WT and *ubp8*Δ strains grown at logarithmic (Log Phase) and stationary phases (Plateau). According to the scheme in log phase cells express both red and green markers in contrast in stationary phase (plateau), while WT shows only red expressed in the vacuole; in the absence of Ubp8 the reporter is not properly localized inside vacuoles indicating defects in the mitophagy progression.

**Figure 4 cells-11-01682-f004:**
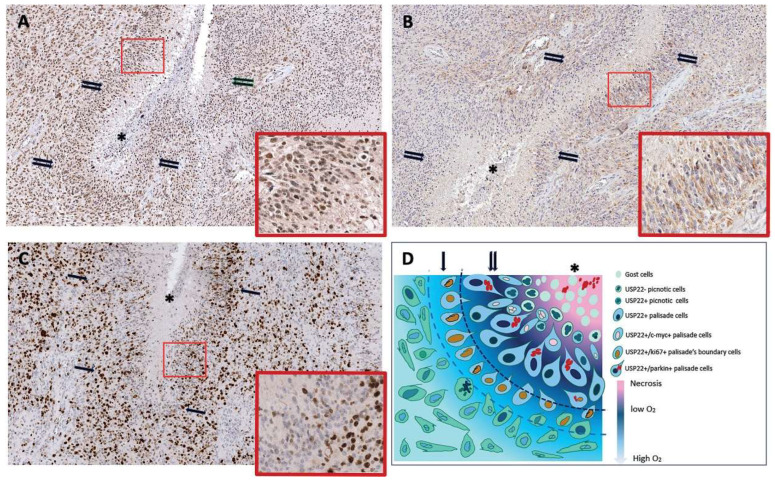
GBM pseudo-palisade immunostaining. (**A**) Usp 22 positive cells highlight the pseudo-palisade layer surrounding the necrosis area (asterisks) with strong staining of the nuclei (double arrows); bottom right panel an enlargement showing the layering of the cells surrounding the necrosis. (**B**) Anti-Parkin staining shows a distribution similar to Up22, strong staining of the cells cytoplasm is visible in the bottom right panel (**C**) Anti-Ki67 positive cells are concentrated in the outer layers of the pseudo-palisade while the internal layers are negative. (**A**–**C**) Immunoperoxidase staining, 100× enlargement. Panels: 400× enlargement. (**D**) Color scheme of pseudo-palisade cell reactivity. Single arrows indicates the pseudopalisade’s boundary cells, double arrows palisade cells.

**Table 1 cells-11-01682-t001:** Saccharomyces cerevisiae strains used.

Strain	Genotype	Source
**W303**	*MATa ade2–1 trp1–1 leu2–3112 his3–11,15 ura3 can1–100*	[29]
**YVDL09**	*MATa ade2–1 trp1–1 leu2–3112 his3–11,15 ura3 can1–100 +* pVT100U-mtGFP	This study
**W303-rho^0^**	*MATa ade2–1 trp1–1 leu2–3112 his3–11,15 ura3 can1–100. rho^0^*	This study
**YVDL11**	*MATa ade2–1 trp1–1 leu2–3112 his3–11,15 ura3 can1–100. rho^0^ +* pVT100U-mtGFP	This study
**YPO4**	*MATa ade2–1 trp1–1 leu2–3112 his3–11,15 ura3 can1–100 gcn5* *::KanMX4*	[29]
**YVDL12**	*MATa ade2–1 trp1–1 leu2–3112 his3–11,15 ura3 can1–100 gcn5**::KanMX4 +* pVT100U-mtGFP	This study
**YV01**	*MATa ade2–1 trp1–1 leu2–3112 his3–11,15 ura3 can1–100 gcn5* *::KanMX4*	This study
**YVDL13**	*MATa ade2–1 trp1–1 leu2–3112 his3–11,15 ura3 can1–100 gcn5**::KanMX4 +* pVT100U-mtGFP	This study
**YFT21**	*MATa ade2–1 trp1–1 leu2–3112 his3–11,15 ura3 can1–100 ubp8::His3MX6*	[29]
**YVDL14**	*MATa ade2–1 trp1–1 leu2–3112 his3–11,15 ura3 can1–100 ubp8::His3MX6 +* pVT100U-mtGFP	This study
**YV02**	*MATa ade2–1 trp1–1 leu2–3112 his3–11,15 ura3 can1–100 ubp8::His3MX6*	This study
**YVDL15**	*MATa ade2–1 trp1–1 leu2–3112 his3–11,15 ura3 can1–100 ubp8::His3MX6 +* pVT100U-mtGFP	This study
**Ros01**	*MATa ade2–1 trp1–1 leu2–3112 his3–11,15 ura3 can1–100* *+ pAS1NBmRosellaI*	This study
**Ros02**	*MATa ade2–1 trp1–1 leu2–3112 his3–11,15 ura3 can1–100 gcn5* *::KanMX4 + pAS1NBmRosellaI*	This study
**Ros03**	*MATa ade2–1 trp1–1 leu2–3112 his3–11,15 ura3 can1–100 ubp8::His3MX6 + pAS1NBmRosellaI*	This study

**Table 2 cells-11-01682-t002:** Histochemical expression of USP22 ubiquitin protease in primary glioblastomas. USP22 expression is given as sum of staining intensity score plus percentage score. Staining intensity score: 0 = negative or very faint nuclear staining, 1 = weak nuclear staining, 2 = strong nuclear and cytoplasm staining. Percentage score: 0 = 5–20%, 1 = 21–60%, 2 = 61–100%.

Total No. of Cases	Cases of Glioblastomas Expressing USP22	Cases with Palisade Necrosis Cells Over-Expressing USP22	Cases with USP22 Comparable Expression in Vital Tumor Cells and Palisade Necrosis Cells	Cases with Palisade Necrosis Cells Ipo-Expressing USP22
**No. of cases**	30/30	19/30	9/30	1/30
**USP22 Mean** **Expression + SD**	2 ± 0.643	2.947 ± 0.229	2.363 ± 0.809	
**Test T**		6.94706 × 10^−13^ *p* < 0		

## Data Availability

The data presented in this study are available on request from the corresponding author.

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
