# Peer review of "Role of yUbp8 in Mitochondria and Hypoxia Entangles the Finding of Human Ortholog Usp22 in the Glioblastoma Pseudo-Palisade Microlayer"

_cells, 2022, doi:10.3390/cells11101682_

Round 1

Reviewer 1 Report

The manuscript by De Luca reports an interesting set of findings on the role of the yeast Gcn5 and Ubp8 in the regulation of respiration. The authors went further on to find that the human Ubp8 ortholog, USP22, can be upregulated in the pseudo-palisading area of GBM, suggesting a role for USP22 in GBM pathology.

Overall, the study was well designed and executed with appropriate controls. The Introduction was well written to lead the readers to the study subject. I find the yeast results are particularly intriguing. However, the link between the yeast results and the human data appears to be rather insufficiently presented. Having extended data on the role for mitophagy components in GBM would make the study much more convincing.

  1. Do the authors have any speculations on why USP22 expression can be upregulated in the GBM pseudo-palisade area?
  2. Can the authors provide us with additional expression data for other components of mitochondria or mitophagy, e.g., Gcn5 ortholog, ubiquitin, etc, using the GBM samples?
  3. In line 13, KAT and DUB should be spelled out.
  4. In Figure 1B, can these lines be compared with a statistical testing?
  5. Images on Figure 4 do not have scale bars. Can the authors show scale bars?
  6. The authors should revise the text carefully. There remain many areas to improve the text in grama and style. Reading the current form of the text is not smooth.

Author Response

Overall, the study was well designed and executed with appropriate controls. The Introduction was well written to lead the readers to the study subject. I find the yeast results are particularly intriguing. However, the link between the yeast results and the human data appears to be rather insufficiently presented. Having extended data on the role for mitophagy components in GBM would make the study much more convincing.

Do the authors have any speculations on why USP22 expression can be upregulated in the GBM pseudo-palisade area? 

Thank you for this challenging question. It is generally known that the cellular responses caused by hypoxia are mainly related to the transcriptional regulation of downstream genes mediated by HIFs (HIF-1α). In a series of studies on GB stem cells development and maintenance it was demonstrated that HIF-1α directly regulates the expression of USP22. Furthermore, using multi-color immunohistochemical analysis of glioblastoma tissue samples they found that HIF-1α-positive cells are characteristically located closer to the necrotic area, thus suggesting the direct early stimulation of USP22. In our opinion and based on literature we think that the activation of USP22 in the pseudo –palisade cells, should have a role in recruiting factors involved in generation of new blood vessels into these very anoxic tissues area. Koutelo and colleagues showed that, in mice, deletion of Usp22 affected the ability to form vessels leading and failure to form proper vascular interactions.

The topic and relative references have been added to the discussion (ref 51 and  52)

Can the authors provide us with additional expression data for other components of mitochondria or mitophagy, e.g., Gcn5 ortholog, ubiquitin, etc, using the GBM samples?

We stained some of the GB sections with anti KATA2/GCN5 (LSBio rabbit anti human LS-B13327), anti-ubiquitin (P37) antibody #58395 (Cell Signaling technology) and BAX (LSBio Monoclonal Mouse anti Human LS B2510).  The expression of GCN5 resulted very faint and diffuse to all the neoplastic cells, whereas the expression of ubiquitin(p37) and BAX was very intense but also these antibodies stained all the neoplastic cells independently from their location in the pseudo-palisades or in diffuse and perivascular growth.
While immuno-histochemistry can be indicative of mitochondrial damage, it is not appropriate to investigate the expression of its components. This task will have to be carried out with other methods, e.g. mRNA and protein extracted from. Pseudo-palisade by laser-microcapture. 

In line 13, KAT and DUB should be spelled out.
Spelled out.

In Figure 1B, can these lines be compared with a statistical testing?    

The experiment was thought as a control based on lack of growth on respiratory medium (gly3%). It was performed twice (added in material and methods) with exactly the same result.

Images on Figure 4 do not have scale bars. Can the authors show scale bars?   

Scale bars added

The authors should revise the text carefully. There remain many areas to improve the text in grama and style. Reading the current form of the text is not smooth.

The whole manuscript has been reviewed and reedited for smoothing by an English mother tongue.

Reviewer 2 Report

In their manuscript, De Luca and colleagues focus on investigating the role of the deubiquitinating enzyme yUbp8 in mitochondria and hypoxia. Notably, they show that yUbp8 is required to maintain mitochondrial function under hypoxia. They further extended their study to the human homologue, Usp22, in cancerous tissues and suggested a conserved function.

The study is interesting and brings new knowledge. However, some aspects may be investigated further to reinforce and fully support the authors' conclusions.

  1. Experiments in yeast would require complementation by re-expressing Ubp8. Is the Ubp8 loss-of-function effect on the mitochondria (localization – Fig 1C – and mito activity – Fig 3C) linked to its catalytic activity? It would be great to complement also with a catalytic inactive Ubp8.
  2. Does yUbp8 also colocalize to mitochondria? It would be interesting to check colocalization (or expression pattern) between mito-GFP and yUbp8.
  3. Is yUbp8 expression altered in WT/ rho strains upon hypoxic conditions?
  4. Figure 3: quantification required.
  5. Figure 4: it is impossible to conclude that there is colocalization between Parkin and USP22 from those micrographs as there is no co-staining. Proper co-staining experiments should be done to support the statement.

Author Response

Comments and Suggestions for Authors

In their manuscript, De Luca and colleagues focus on investigating the role of the deubiquitinating enzyme yUbp8 in mitochondria and hypoxia. Notably, they show that yUbp8 is required to maintain mitochondrial function under hypoxia. They further extended their study to the human homologue, Usp22, in cancerous tissues and suggested a conserved function.

The study is interesting and brings new knowledge. However, some aspects may be investigated further to reinforce and fully support the authors' conclusions.

Experiments in yeast would require complementation by re-expressing Ubp8     

In this study we have reported findings and performed our analysis in S.cerevisiae strain disrupted in UBP8 and the full deletion of the gene was confirmed by direct colony-PCR. Yeast reverse genetics is and has been used for decades to study the effects of specific deletion of single genes as one of the most powerful approach in the study of gene function.

In our opinion the complementation test is not determinant because we were interested in the direct effects determined by the comparison between the lack of UBP8 gene in the New and Old disrupted strain.

Moreover, in New deleted strain we observed a progressive loss of functions in time that was only fixed in ubp8D-Old strain, therefore, the complementation test will only reinstall the same phenotypes observed in New strain.

Finally, Ubp8 is a subunit of SAGA multiproteic complex, for this reason, the expression of Ubp8 in a complementation test might not be at physiological levels adding additional, uncontrolled and undesired effects on the overall functionality of the whole SAGA complex.

Is the Ubp8 loss-of-function effect on the mitochondria (localization – Fig 1C – and mito activity – Fig 3C) linked to its catalytic activity? It would be great to complement also with a catalytic inactive Ubp8

The question that you posed is very interesting, however we think that following the deletion of UBP8 in both new and old strain the activity must be lost immediately and not after many generations, it follows that only the activation of alternative metabolic pathways might be responsible and explains the observed growth under anoxic conditions.

Does yUbp8 also colocalize to mitochondria? It would be interesting to check colocalization (or expression pattern) between mito-GFP and yUbp8 

Yes, it does, we published this result in 2018 and reported related references [28-30] recalling these findings (l. 53):  

…. “Consistently, we reported the role of KAT-Gcn5 and ubiquitin protease Ubp8 in yeast respiration and their localization into mitochondria as single protein”

Is yUbp8 expression altered in WT/ rho strains upon hypoxic conditions?

We would like to recall that the rho- strain was used as a control since it lacks of mitochondria and therefore unable to respire and grow on respiratory carbon sources. We think that the level of expression of Ubp8 in rho- strain can’t restore any respiratory activity in absence of mitochondria. 

Figure 3: quantification required.

We agree with the Reviewer and added quantification data of Rosella experiment in the text: lines, 296-301

“The log phase WT and ubp8Δ cells showed the typical distribution of yeast mitochondrial reticulum stained in red/green. In the log/stationary phase, the vacuoles of WT strain showed % 69,51 ± 1,80 cells stained in red, indicating the accumulation of the Rosella red fluorescent proteins inside vacuoles. Interestingly, the vacuolar localization of Rosella was less defined in ubp8Δ cells 9,16 ± 1,11 meaning that mitophagy was not so efficiently activated in the cells.”

Figure 4: it is impossible to conclude that there is colocalization between Parkin and USP22 from those micrographs as there is no co-staining.

As shown in figure 4B panel enlargements, about half of the elongated cells of the pseudo-palisade present small granular clumps of brown stain in the cytoplasm, based on this evidence we were confident to state that this particular expression of Parkin colocalize with USP22+ cells in the pseudo-palisade.

Since the figure 5B was taken on the same palisade on sequential sections, though with a different orientation of the sections on the slides, and figure 4A (panel enlargement) shows that almost all the nuclei are USP22 positive we can assume that some of these cells also express Parkin in the cytoplasm.

We agree that co-staining will give a more elegant and precise demonstration that USP22 and Parkin are expressed by the same cells and during the revision time we have tried this co-staining experiment. Unfortunately, being our antibodies mouse-anti human and Parkin expression very low, notwithstanding all the suggested precautions only the prevalent expression of USP22 could be revealed clearly.    

Reviewer 3 Report

The paper is very good and deals with an extremely important subject. The research described in this paper is an important step in further research into the pathophysiology of glioblastoma. I consider relevant the involvement of mitochondria in the pathology of glioblastoma. The paper is written concisely and clearly presents the data that support the result.

1 - the article reports the role of Ubp8 in hypoxic conditions and the authors presented how it allows growth, with altered mitophagy

2 - they translated what they discovered in the yeast to human ortholog gene Usp22 in glioblastoma tumor specimens

3 - they demonstrated the presence and overexpression of Usp22 in the pseudo-palisade tissue around the necrosis area of the tumor

4 - they correlated the data with mitochondrial functions in GBM, which explains a pattern of proliferation and invasion of glioblastoma cancer cells - an important and worthwhile idea

5 - experiments and results are clearly described - can be reproduced

6 - the conclusion is clear and useful, indicating new lines of research

Author Response

Comments and Suggestions for Authors

The paper is very good and deals with an extremely important subject. The research described in this paper is an important step in further research into the pathophysiology of glioblastoma. I consider relevant the involvement of mitochondria in the pathology of glioblastoma. The paper is written concisely and clearly presents the data that support the result.

1 - the article reports the role of Ubp8 in hypoxic conditions and the authors presented how it allows growth, with altered mitophagy

2 - they translated what they discovered in the yeast to human ortholog gene Usp22 in glioblastoma tumor specimens

3 - they demonstrated the presence and overexpression of Usp22 in the pseudo-palisade tissue around the necrosis area of the tumor

4 - they correlated the data with mitochondrial functions in GBM, which explains a pattern of proliferation and invasion of glioblastoma cancer cells - an important and worthwhile idea 

5 - experiments and results are clearly described - can be reproduced

6 - the conclusion is clear and useful, indicating new lines of research

We thank Rev. 3 for appreciation.

Round 2

Reviewer 1 Report

The revised manuscript has addressed the issues that I raised before. The manuscript now appears much clearer and reads well. I applaud the authors for their efforts to improve their manuscript.

Reviewer 2 Report

In the revised version of their manuscript and joined letter, the authors have provided sensible answers and clarified some points.

Therefore, I recommend this article for publication in its present form.